# Applied aerial spectroscopy: A case study on remote sensing of an ancient and semi-natural woodland

**Shara Ahmed[1], Catherine E. Nicholson[1], Paul Muto[2], Justin J. Perry[1], John R. Dean[1]***

**1** Department of Applied Sciences, Northumbria University, Ellison Building, Newcastle upon Tyne, United Kingdom, **2** Natural England, Lancaster House, Hampshire Court, Newcastle upon Tyne, United Kingdom

* john.dean@northumbria.ac.uk

**Data Availability Statement:** All relevant data are within the manuscript and its Supporting Information files.

## Abstract

An area of ancient and semi-natural woodland (ASNW) has been investigated by applied aerial spectroscopy using an unmanned aerial vehicle (UAV) with multispectral image (MSI) camera. A novel normalised difference spectral index (NDSI) algorithm was developed using principal component analysis (PCA). This novel NDSI was then combined with a simple segmentation method of thresholding and applied for the identification of native tree species as well as the overall health of the woodland. Using this new approach allowed the identification of trees at canopy level, across 7.4 hectares (73,934 $m^2$) of ASNW, as oak (53%), silver birch (37%), empty space (9%) and dead trees (1%). This UAV derived data was corroborated, for its accuracy, by a statistically valid ground-level field study that identified oak (47%), silver birch (46%) and dead trees (7.4%). This simple innovative approach, using a low-cost multirotor UAV with MSI camera, is both rapid to deploy, was flown around 100 m above ground level, provides useable high resolution (5.3 cm / pixel) data within 22 mins that can be interrogated using readily available PC-based software to identify tree species. In addition, it provides an overall oversight of woodland health and has the potential to inform a future woodland regeneration strategy.

## 1. Introduction

Remote sensing is the scanning of an object or phenomena from a distance by a high-flying aircraft or satellite to obtain relevant information [1]. However, the use of satellite-based platforms for monitoring has a few limitations. Typically, the spatial and temporal resolution of satellites are relatively coarse e.g. a 1 km resolution coupled with a multiday revisit cycle [1]. In addition, the quality of satellite images is susceptible to weather, such as, cloud coverage and atmospheric absorption which needs to be corrected [2, 3]. Due to the drawbacks of satellite platforms, a remote sensing technique using an unmanned aerial vehicle (UAV) has increasingly gained interest. A UAV can capture images with high spatial resolution in the scale of centimetres (1 to 50 cm) and with high temporal resolution allowing capture of images

**Funding:** The authors received no specific funding for this work.

**Competing interests:** The authors have declared that no competing interests exist.

multiple times each day. Also, UAVs are less susceptible to the weather, specifically cloud cover, due to their significantly lower flight altitude e.g. 100 m; UAVs can fly below low clouds removing any potential barrier from their field of view. In addition, UAV platforms are gradually becoming less expensive to operate and maintain which allows them to be used for a wide range of monitoring applications. UAVs, otherwise known as drones, are aircraft that fly without the need of an onboard pilot. Drones can fly autonomously or with the help of a remote control 'pilot' on the ground. The three main types of drones are the fixed wing, single rotor (small helicopter) and multirotor craft. The recent availability of UAVs with multispectral image cameras allows for their use for applied aerial spectroscopy (AAS). The main type of cameras in UAVs are the RGB (red, green and blue) camera and the multispectral camera (with the RGB spectral channel and NIR channels).

Local Nature Reserve woodlands, in the UK, are protected places for native species of trees and shrubs that provide habitats for numerous species of fungi, invertebrates, birds, mammals and reptiles that all contribute to provide a natural, functioning, and balanced ecosystem [4]. The history of woodlands in the UK can be dated as far back as the last Ice Age (12,000 years ago), where open bare land was colonised with initially juniper, willow, aspen, birch, and Scot's pine and then as the climate improved by hazel, alder, oak, elm and lime [5]. By the Neolithic period (4500-2000 BC) the UK was covered by wild woodland, however this situation changed considerably by the Iron Age (750 BC- 40 AD) were 50% of the woodland was cleared due to agricultural activity, and by the 20th century it had reached as low as 5% [6]. Hence, it was after this that a need for conserving woodland was recognised which resulted in coniferous plantations of trees and from 1950 the conservation movement began to protect ancient woodland as nature reserve sites [7]. The term ancient woodland has been used to describe woodland that dates from 1600 AD, however some of these woodlands have been modified by human activity and so are often referred to as ancient and semi-natural woodland (ASNW) [5]. At present, the UK is covered by 13% of woodland, of which half of this is coniferous plantation and only 1.2% is ASNW [7]. In comparison with some European countries, the UK was slow to acknowledge the need to conserve nature reserve woodlands resulting in one of the least-wooded countries. However, on-going activities continue to conserve and protect the remaining small-scale woodlands which remain under threat from urban development [5]. It is therefore an important cultural and social responsibility to protect the destruction of the woodlands and to restore and create natural ecosystems that are diverse and rich in wildlife. There are approximately 340,000 ha of ancient woodland in England, defined as sites with continuous woodland cover since 1600 AD. Of this, 200,000 ha is considered semi-natural, that is, it is of natural origin rather than artificially planted [6].

Methods for monitoring native tree species in woodland has traditionally involved field-based assessment methods to assess trees below the canopy level [8]. However, field assessment methods are time consuming and costly, hence the interest in the use of remote-sensing technology for habitat monitoring [9, 10]. Unlike field assessment methods, remote sensing techniques can provide full coverage at different resolution with respect to different spectral bands, space and time thereby allowing forest modelling of species recognition [11]. Remote sensing is a non-destructive technique which uses a diverse array of sensors and platforms to collect information from above the Earth's surface without having contact with the object. Sensors range from RGB, multispectral, hyperspectral, light detecting and ranging (LiDAR), synthetic aperture radar (SAR) and optical which are mounted on a range of platforms from airborne, spaceborne, to unmanned aerial vehicle (UAV) [12].

Traditional airborne remote sensing platforms using manned aircraft flying at an altitude ranging from 50 m to 3000 m using LiDAR and aerial laser scanner (ALS) have been used to estimate the heights of tree which contributed to information for tree species classification

[13–16]. Further, considering the spaceborne remote sensing platforms the most widely used for tree detection and delineation are Phase array type L-band Synthetic Aperture Radar (PAL-SAR), Google Earth, WorldView-2, WorldView-3 and Quickbird [17–24]. Furthermore, most of this satellite imagery is categorised as multispectral imaging using between 2 and 15 spectral bands and with high spatial resolution (varying between 46 cm and 184 cm for Worldwide-2 and 31 cm and 124 cm for WorldView-3). However, these traditional remote sensing platforms are unsuitable for regional or local forestry purposes as they are expensive and have an increased temporal resolution which ultimately delays regular time series monitoring. UAV platforms, on the other hand, can capture images with exceptional spatial resolution (centimetre scale) and with the high temporal resolution allowing multiple images to be captured each day [12]. UAVs are also less susceptible to weather as they can fly below low-lying clouds, removing any potential obstruction from the field of view. Their low-cost of operation and capital cost allows them to be used frequently [25]. The two main platforms used in UAVs are fixed wing and multirotor vehicles. Although fixed-wing vehicles have been the preferred option by most researchers, they have several disadvantages when compared to multirotor vehicles; fixed-wing vehicles can only move forward, are difficult to land and cannot hover in the air [12]. On the other hand, multirotor vehicles can hover, are easy to control due to their stability and can take off and land vertically [25]. The main disadvantage of UAVs is inflicted by battery duration; however, the advantages of UAVs far outweigh this drawback in comparison to other remote sensing platforms and can be used as a valuable tool in monitoring and mapping tree species [26]. The assessment of monitoring tree species by UAV offers a wide range of sensors of which the most common are RGB, multispectral and hyperspectral. The RGB sensors are insufficient to monitor tree species types due to their low spectral resolution while hyperspectral sensors are expensive and unsuitable for long term monitoring of woodland [8, 9]. Currently there is an exponential growth in the usage of hyperspectral sensors for species recognition [27–30]. However, hyperspectral imaging utilizes hundreds of contagious spectral bands to acquire images which makes them computationally heavy, are time consuming and requires expertise to interpret the data.

The most promising UAV sensors are multispectral which are cost efficient and capable of capturing images across different spectral channels of the visible and NIR regions with increased spectral resolution applicable for woodland monitoring [31]. Among the noteworthy recent research studies using multispectral UAV imaging for species recognition were areas where biodiversity of tree species was high, such as, Canada, Italy, Nepal, and Costa Rica where machine learning methods using object based-image analysis methods (OBIA) were used to determine target species. In this manner, classification of target plant species of aspen, white birch, sugar maple and red maple in Canada and trees and shrub species along the Himalayan ecotone in Nepal were established by multi resolution and random forest methods [32, 33]. Further, classification of various plant species in a plant nursery in Italy were classified by using maximum likelihood classifiers [34]. Also, Yaney-Keller et al., 2019 used a spatial resolution of 100 cm/pixel to discriminate seven abundant mangrove species in Costa Rica using the Normalised Difference Vegetation Index (NDVI) which were then further classified by support vector machine learning [35]. These classifiers using support vector machine learning, random forest, and maximum likelihood are examples of supervised machine learning algorithms which offer high accuracy at determining target tree species. However, these algorithms require extensive computing time to train the data set. In addition, the developed and built algorithms still had limitations on their applicability to be deployed to similar datasets without further modification. Therefore, less time-consuming algorithms, which are user-friendly, accurate and less complicated are still required to monitor tree species, especially in situations where frequent analysis of tree species are required. Hence, image processing methods which

use algorithms, such as, the normalised difference spectral index (NDSI) can be more easily derived by utilising information from multispectral images which enhance and distinguish spectral features in tree species classification. The most used spectral index or vegetation index (NDVI) are often used to monitor the growth and health of crops [36]. Other vegetation indices, such as, the optimised soil adjusted vegetation index (OSAVI) is used for crop management whereas the biomass estimation and enhanced vegetation index (EVI) are used to quantify the vegetation greenness [37]. So far, each developed vegetation index has been used specifically for a particular purpose. Therefore, it is proposed that a more suitable approach is to use the information from multispectral bands to derive a more robust NDSI which can determine tree species based on their varying amounts of pigments (i.e. chlorophyll and carotenoids). Furthermore, an enhanced sensitivity of individual multispectral images (before deriving a NDSI) can be done by performing Principal Component Analysis (PCA). PCA is a multivariate statistical method often applied in image analysis and classification to reduce the amount of redundant information. The approach then provides the most important information in an image which can be used to differentiate between different scene elements (e.g. different species of trees) [38]. The discriminated tree species derived from a specific NDSI can be further interrogated by the simpler segmentation method of thresholding. Thresholding is a pixel-based method which segments regions in an image by intensity values. Unlike OBIA methods, the thresholding method only considers the spectral features of an image. Therefore, if the tree species are discriminated sufficiently by NDSI, thresholding is a useful method for classifying tree species.

Our aim is to use a UAV, with multispectral image sensors to monitor an ASNW (i.e., Priestclose Wood) with simpler, but robust and appropriate digital software manipulation to (a) perform PCA classification on multispectral images to help derive a suitable NDSI which allows discrimination of native tree species, (b) to segment the discriminated native tree species by a thresholding method, (c) to compare the performance of the established NDVI algorithm with the new NDSI algorithm approach to discriminate between the native tree species, (d) to investigate woodland coverage at canopy level with respect to the overall health of the woodland, (e) to assess the accuracy of our UAV MSI data findings with field study data obtained at ground level, and (f) to contribute to the evolving field of precision woodland management.

## 2. Methodology

### 2.1 Sampling site

Priestclose Wood is an ancient woodland located on the eastern edge of the conurbation of Prudhoe (population 11700, based on 2011 census [39], Fig 1(A)) located in the Tyne Valley, south Northumberland, northeast England (Fig 1(B)) (OS map reference NZ 107 627). The total area of the woodland is 19.35 hectares (193,500 m$^2$), of which 15.26 hectares (152,600 m$^2$) is categorised as ancient and semi-natural and managed as a Local Nature Reserve by the Northumberland Wildlife Trust (NWT) [40] (Fig 1(A)). Permission to fly the drone over the ASNW was provided by Mr Geoff Dobbins, Senior Estates Officer (Reserves), Northumberland Wildlife Trust Ltd. The wood is predominantly a mixture of Pedunculate and Sessile oak (*Quercus robur* and *Quercus petraea*) and silver birch (*Betula pendula*), although rowan (*Sorbus aucuparia*), hazel (*Corylus avellana*), holly (*Ilex aquifolium*), wych elm (*Ulmus glabra*), sycamore (*Acer pseudoplatanus*) and some ash (*Fraxinus excelsior*) is also present.

Assessed with the ground vegetation, the woodland best fits the W10 (*Quercus robur - Pteridium aquilinum - Rubus fruticosus* woodland) British National Vegetation Classification (NVC) community. There is also an area of wet woodland on the northern boundary

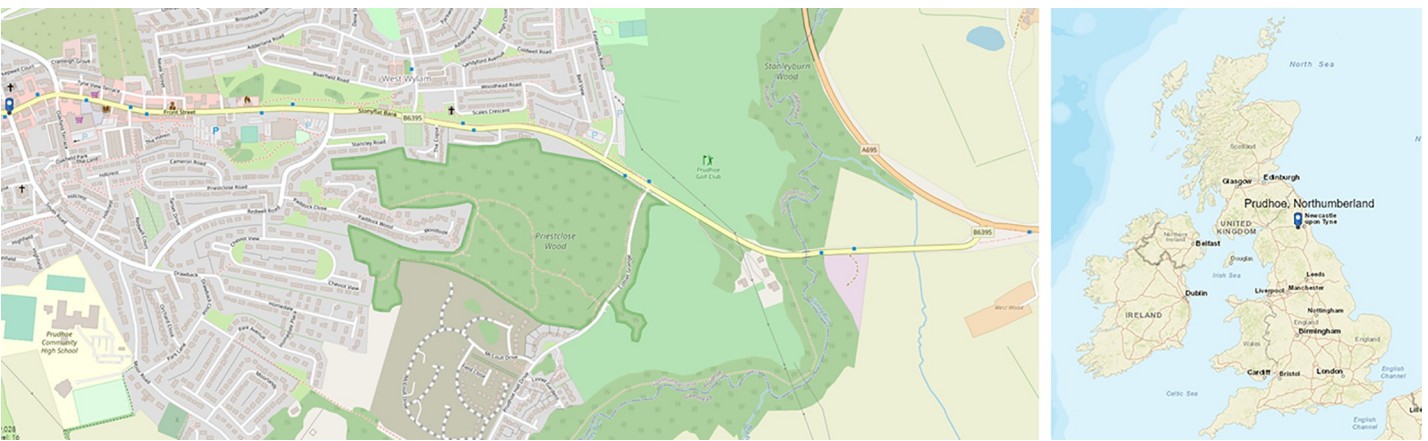

**Fig 1. Location of (a) Priestclose Wood, and (b) Prudhoe, Northumberland, UK.** [USGS National Map Viewer; USGS National Map Viewer]

(adjoining the B6395 road) consisting primarily of downy birch (*Betula pubescens*) and willow species with a ground flora grading from a fen-like W2 (*Salix cinerea - Betula pubescens - Phragmites australis* woodland community) to a bog-like W4c (*Betula pubescens-Molinia caerulea* woodland, Sphagnum sub-community). The eastern edge of the wood, adjoining Prudhoe Hall Drive, is dominated at canopy level by oak trees, but at lower level by rhododendron bushes. The southern boundary, adjoining Cottier Grange housing estate, is a planted boundary added to the ancient woodland in the 1890s, featuring both Scots (*Pinus sylvestris*) and Austrian Pine (*Pinus nigra*) as well as Norway maple (*Acer platanoides*). Sycamore (*Acer pseudoplatanus*) is also common along the boundaries of the woods. In the 1930s, a fire affected a portion of the woodland, resulting in the removal of the affected oak trees to a local sawmill resulting in natural regeneration predominantly of silver birch. The removal of trees for firewood also took place during the 1970s, resulting in further areas of natural regeneration. The central area of the woodland has areas of standing dead trees which can provide a crucial habitat for biodiversity and an open space for tree regeneration [40]. The chosen area to investigate, an approximate rectangle of 7.4 hectares (73,934 m$^2$), bordered the streets located between the B6395, Woodside, Cottier Grange housing development and Prudhoe Hall Drive (Fig 2). This part of the ASNW consists mainly of oak and silver birch with an understory of holly and hazel.

## 2.2 Unmanned aerial vehicle

A multirotor UAV (DJI Phantom 4, supplied by Colena Ltd., North Shields, UK) was used with multispectral camera. The multispectral image camera operates with a 5 camera-array covering the blue (450 ± 16 nm), green (560 ± 16 nm), red (650 ± 16 nm), red edge (730 ± 16 nm) and near infrared (840 ± 26 nm) spectra with an additional camera that can also provide live images in RGB (visible) mode as well as in normalised difference vegetation index (NDVI) mode. All cameras are stabilised with a 3-axis gimbal. In all cases the camera was angled perpendicular to the ground, with data capture occurring in hover & capture mode. The UAV speed was 5.0 m/s and an average height of 100 m. All flights were recorded with a resolution of 5.3 cm/px, a front overlap ratio of 75%, a side overlap ratio of 60% and a course angle of 90˚. Specific weather conditions relating to daytime temperature during flight, wind speed and direction (both recorded using a hand-held anemometer (Benetech® GM816, Amazon UK)), and UAV pilot anecdotal observation on cloud coverage, are identified with specific dated

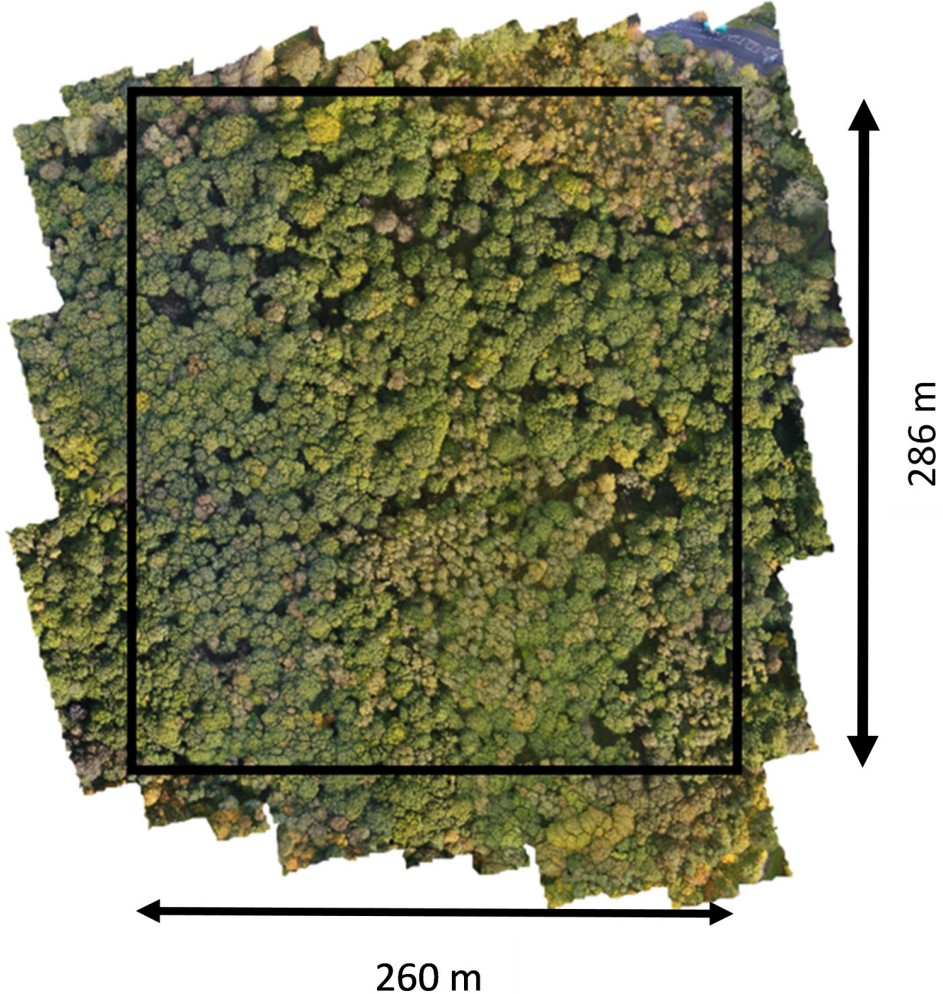

**Fig 2. Visible mode software stitched image of Priestclose Wood with actual area used for analysis.**

data. Data from the central region was recorded using 178 waypoints, over 13 lines, with a total flight length of 3921 m. Flight data was recorded over the period 17 September – 14 October 2020 between the hours of 10:30 and 17:10, respectively, with average flight times around 22 minutes. The total number of images collected, per flight, was 1076.

## 2.3 Field data analysis

Field data was obtained at ground level over the period of late 2020 / early 2021. Field data in terms of photographic evidence was obtained illustrating the typical canopy coverage (Fig 3 (A)), visual identification of dead trees (Fig 3(B)) and other low level plant growth i.e. holly bushes (Fig 3(C)). Field sampling to identify trees was done over the weekend of 30/31 January 2021 to identify the number of oak and silver birch trees (by their bark) present at canopy level as well as the number of dead trees both above and below canopy level. A total field site of 10,800 m$^2$ was covered, by foot by two independent tree surveyors. Subsequently (16 February 2021) the average tree height of silver birch and oak trees in the field survey area was done, by two independent tree surveyors, using a hand-held clinometer and simple algebra.

**(a)**

**(b)** **(c)**

[camera: a Nikon D3500 body with a Nikon DX VR AF-P Nikkor 18-55 mm 1:3.5-5.6G (operated in automode); photographs (a) taken on 1 October 2020, (b) and (c) taken on 31 January 2021].

**Fig 3. Example ground level photographs of (a) central region (canopy coverage), (b) dead tree, and (c) holly bush.**

### 2.4 UAV data analysis

**2.4.1 Photogrammetric processing.** From the images taken by the multispectral UAV, an orthomosaic image was produced by using Agisoft Metashape Professional (64 bit) software v.1.7.1 (Agisoft LLC, St. Petersburg, Russia). Agisoft parameters for photogrammetric processing were performed as follows. Initially, the individual images were aligned with medium accuracy (key point limit: 40,000 and tie point limit: 40,00). Then, a dense point cloud was built with low quality and aggressive depth filtering. Afterwards a mesh model was built with the following parameters; surface type: height field, source data: sparse cloud, polygon count: high, advanced interpolation: enabled and calculated vertex colours: checked. Finally, the orthomosaic image was built and saved as a tif file. The software provides an automated image processing sequence to align multiple individual images that can be stitched together to build an orthomosaic image, also known as an aerial image (Fig 2). Fig 2 indicates the area used for UAV data analysis.

**2.4.2 Image processing and data analysis.** Further image processing and implementation of algorithms, such as, PCA and vegetation indices (VI) on the multispectral UAV images were performed using MATLAB v.R2020b (MathsWorks Inc, USA) a programming language software. The workflow of processing data is summarized in Fig 4.

**2.4.3 Normalised Difference Vegetation Index (NDVI).** NDVI is a mathematical formula to determine the growth of vegetation in a particular area and more specifically the health of the vegetation [36]. The NDVI is calculated by the difference between reflected energy in the NIR and red spectral bands normalised by the sum of both NIR and red spectral bands using the equation below:

$$\mathrm{NDVI} = \frac{(\mathrm{NIR} - \mathrm{Red})}{(\mathrm{NIR} + \mathrm{Red})} \tag{1}$$

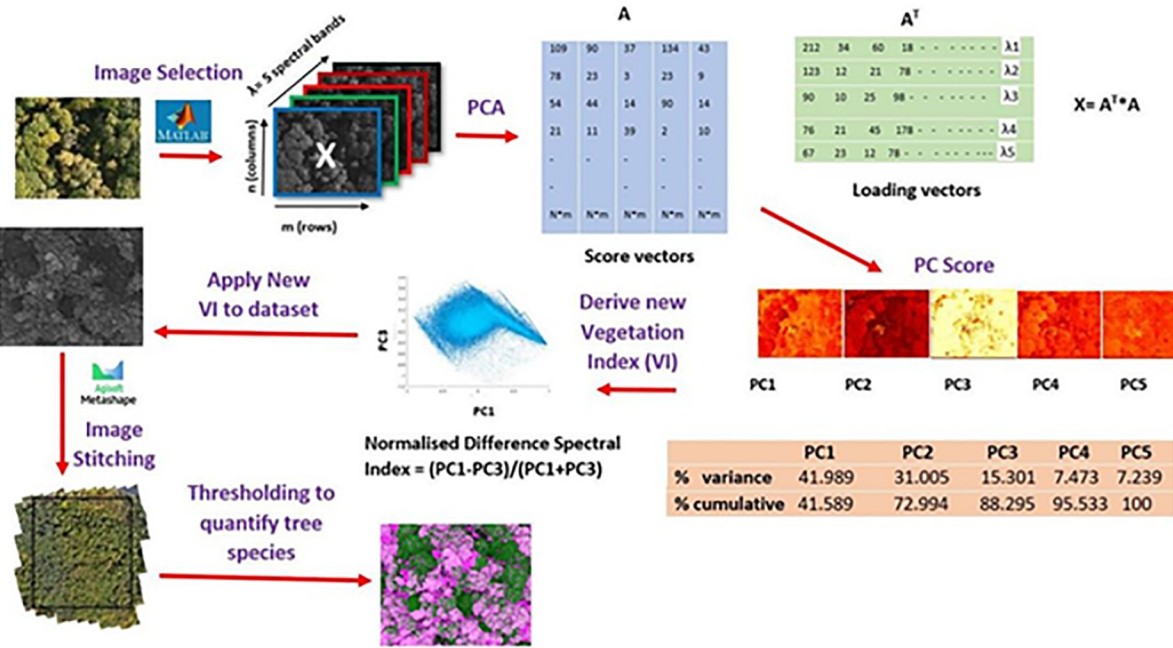

**Fig 4. Data processing workflow for Priestclose Wood.**

The NDVI values range between [−1; + 1]. A pixel value close to 1, indicates a huge increase in the reflectance energy in the NIR spectral band and low reflectance energy in the red spectral band; this corresponds to a greater spectral response of vegetation development and density [41]. Values closer to -1 have a greater reflectance energy in the red spectral band than the NIR spectral band and are considered as water saturated areas [41]. Ideally, NDVI might generate different spectral index values for different types of tree species as the reflectance energy between the NIR and red spectral bands might be different for different types of tree species in an area. Therefore, NDVI images were generated by MATLAB using the red and NIR multispectral images to test its effectiveness in classification of native tree species in the ASNW.

**2.4.4 Data analysis using principal component analysis.** The central area of Priestclose Wood has areas of standing dead trees and native tree species, particularly oak and silver birch trees (Fig 3). PCA was applied to classify the dead and native tree species in the woodland and compare the results with NDVI. From the image data set of the woodland, RGB (visible) images were used to identify the set of images which display the dead trees (Fig 5(A)) and native tree species (Fig 5(B)). Visual observation of the images from the data set was crucial as all the images, from the data set, do not contain dead trees and some images do not have both silver birch and oak trees; additionally, by not randomly selecting an image from the data set would not serve the purpose of this study. In this manner, two multispectral image data sets representing dead trees (Fig 6(A)) and both silver birch and oak tree (Fig 6(B)) were selected. The multispectral images were red, green, blue, red edge and NIR which were then run using MATLAB to perform the PCA which subsequently generated the eigenvectors and % variance. Table 1 shows the features extracted, by PCA, from the multispectral image data set. By using the data obtained for the first three principal component (PC) scores, identifies the % cumulative total for the dead tree images as 86.005 (Table 1(a)) and for oak and silver birch trees as 88.295 (Table 1(b)). This implies that these images retain the most important information that can be used for effective data analysis.

A new spectral index can be calculated using some of these pixel values as follows:

$$\text{Normalised Difference Spectral Index (NDSI)} = (x1 - x2)/(x1 + x2) \qquad (2)$$

This NDSI is the difference between the two spectral bands (x1 and x2) normalised by their sum; it can be used to differentiate scene elements and enhance spectral features that are not

**(a)**                                                                 **(b)**

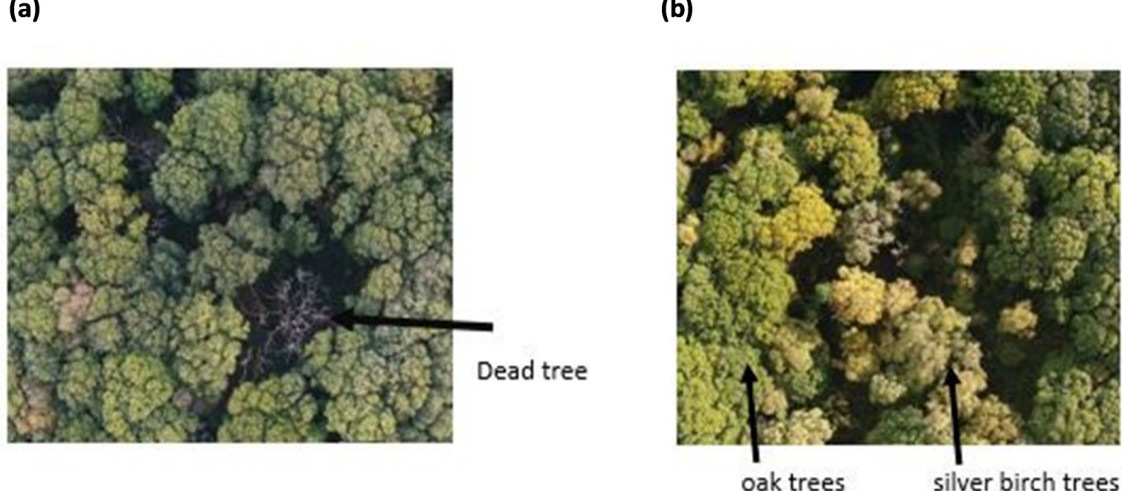

**Fig 5. RGB (visible) images from the data set of the central area of Priestclose Wood (a) Dead trees with a canopy coverage of oak trees (b) Oak and silver birch trees.**

**(a)**

**(b)**

**Fig 6. Multispectral images used from the central area of Priestclose Wood to perform PCA (a) multispectral image data set representing dead trees (b) multispectral image date set representing oak and silver birch trees.** [1 = Blue, 2 = Green, 3 = Red, 4 = DR and 5 = NIR].

visible. In addition, PCA allows a selection of multiple spectral bands thereby providing the option to select and eliminate the appropriate spectral bands. Hence, appropriate PCs were used to derive an NDSI to differentiate the dead trees from the oak and silver birch trees (Fig 7). According to Fig 7, PC1 and PC3 shows a huge difference in spectral responses at different spectral bands in both dead trees and oak and silver birch trees. The difference in spectral response for dead trees (Fig 8A–8C) was found in the green and red spectral bands. For oak and silver birch trees (Fig 8D–8F) a difference in spectral response was observed in all five spectral bands, with a pronounced difference observed in the blue, red and NIR spectral bands. However, comparing the spectral bands in PC1 and PC2 for both dead and oak trees only, a difference was observed in the blue and green spectral bands which was insufficient at extracting useful features; this was because the spectral response of vegetation mainly uses spectral bands from the red to NIR (Fig 9). Hence, a new NDSI was derived using both PC1 and PC3 as below:

$$NDSI = (PC1 - PC3)/(PC1 + PC3) \tag{3}$$

This new spectral index was applied to the ASNW data to generate multispectral images which were used for further data analysis.

**Table 1. Percentage variance of PC1 – PC5 resulting from PCA applied to multispectral images to classify (a) dead trees, and (b) oak and silver birch trees.**

| (a) | PC1 | PC2 | PC3 | PC4 | PC5 |
|---|---|---|---|---|---|
| % variance | 41.470 | 30.758 | 13.777 | 7.473 | 6.522 |
| % cumulative | 41.470 | 72.228 | 86.005 | 93.478 | 100.000 |
| (b) | | | | | |
| % variance | 41.989 | 31.005 | 15.301 | 7.473 | 7.239 |
| % cumulative | 41.989 | 72.994 | 88.295 | 95.533 | 100.000 |

**(a)**

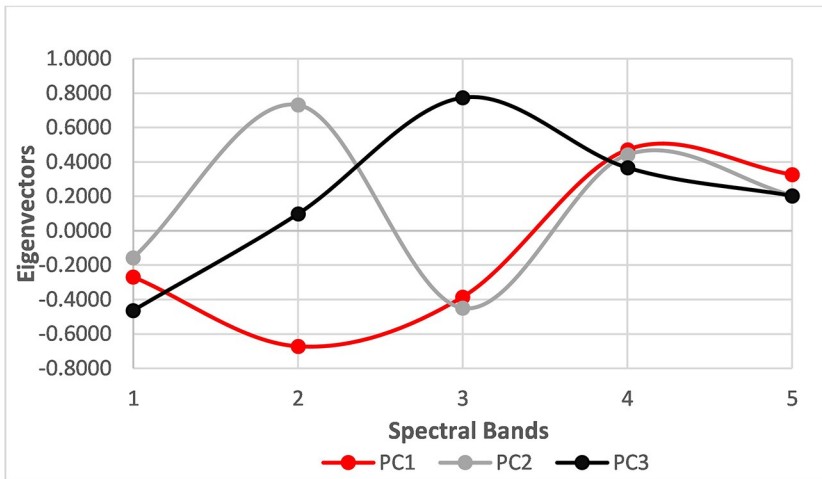

**(b)**

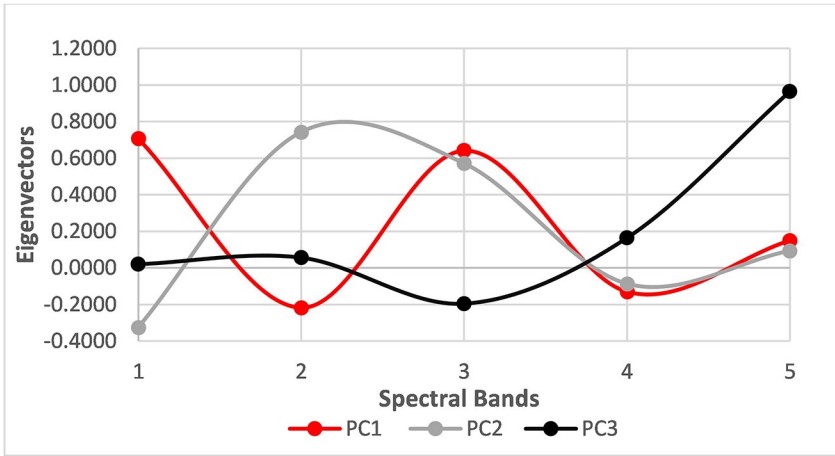

**Fig 7. Eigenvectors indicating the proportion of each spectral band contributing to form each individual PC1, PC2 and PC3 image (a) for dead trees (b) oak and silver birch trees.** [1 = Blue, 2 = Green, 3 = Red, 4 = DR and 5 = NIR].

**2.4.5 Calculation of pixels of native tree species and dead trees.** Images derived from the new NDSI = (PC1 - PC3)/(PC1 + PC3), were stitched using Agisoft Metashape Professional software to build an orthomosaic image. The orthomosaic image which represents the ASNW is shown in Fig 2. The new NDSI could highlight the different spectral responses from the native tree species and dead trees thereby, assisting in classification of tree species. The classified regions of trees by the NDSI algorithm were then segmented by the thresholding method, where the images were converted to a binary image and assigned a different pixel value for oak, silver birch and dead trees.

The total pixel numbers for each area representing oak tree, silver birch tree and dead tree were counted using MATLAB. The individual images composed of the new NDSI were stitched and converted into a grayscale image (S1 and S2 Figs). Interpretation of this grey scale image allowed individual threshold values to be determined for oak (0.35 to 0.38), silver birch (0.44 to 0.48) and dead trees (0.20). It is worth noting that the threshold values for oak and silver birch trees varied slightly over the data collection time period of 4 weeks due to the natural

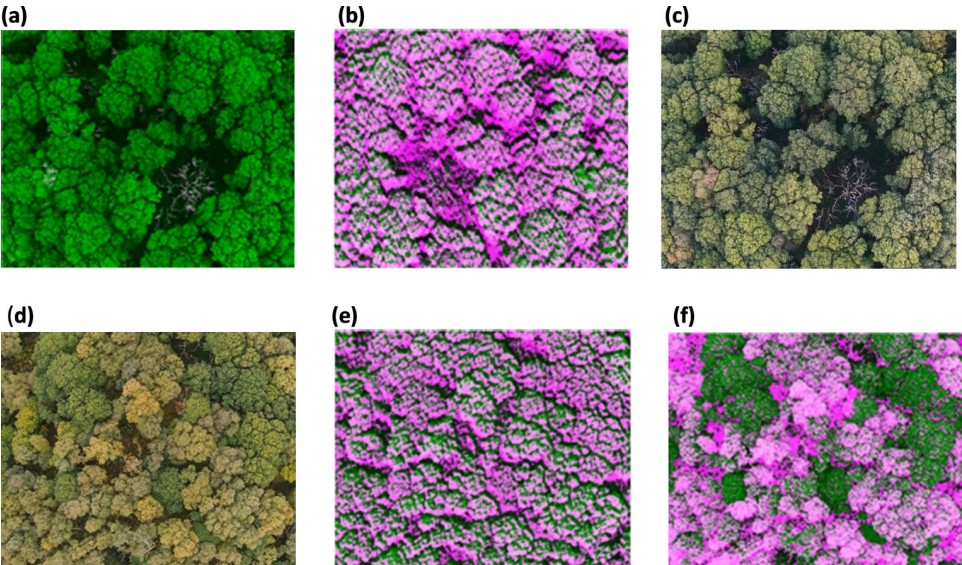

**Fig 8. Thresholding Images derived from NDSI for dead trees (a) RGB image, (b) image derived from (PC1-PC2)/(PC1+PC2) (c) image derived from (PC1-PC3)/(PC1+PC3).** For silver birch and oak trees (d) RGB image (e) image derived from (PC1-PC2)/(PC1+PC2) (f) image derived from (PC1-PC3)/(PC1+PC3).

phenological cycle of the trees (S1 Fig). The total pixel count of each respective area was then multiplied by the resolution of the captured drone images (0.053 m / pixel * 0.053 m / pixel) to represent the area of woodland investigated. The percentage coverage of dead trees and canopy coverage of oak and silver birch trees were calculated by dividing each respective area by the total area of the woodland. To assess the precision of the data, the pixel calculation for the native tree species and dead trees was analysed on 5 separate occasions over 4 weeks in late summer / early autumn.

## 3. Results and discussion

### 3.1 Comparing the performance of classification methods by principal component analysis

The result from the study demonstrates the performance of classification methods of NDVI, PCA applied NDVI and newly derived NDSI from PCA to identify the native tree species of oak and silver birch trees along with dead trees from the ASNW. The new NDSI algorithm derived from PCA allows for classification of native tree species where the dark green coloured regions are identified as oak trees and the light green coloured regions are identified as silver birch trees (Fig 9(B)). However, PCA derived NDSI or NDVI algorithms on their own are inadequate for classification therefore, to enhance the classification, thresholding was executed further to separate the native tree species into segments according to their intensity values.

In this manner, initially, oak, and silver birch trees were identified by NDVI and the PCA applied NDVI algorithms, and afterwards the trees were further classified by threshold seg- mentation. The NDVI images were composed by using the original NIR and red multispectral UAV images. Whereas the PCA applied NDVI images were composed by using NIR and red eigen vectors from PC1 (Table 1(B)). In this case, PC1 consisted of 41% of the total variance implying the high proportion of original information retained in PC1. The NDVI images after the application of the eigen vectors from PC1 (Fig 9(F)) showed a better classification of both oak in the fluorescent green coloured segments and silver birch trees in the purple-coloured

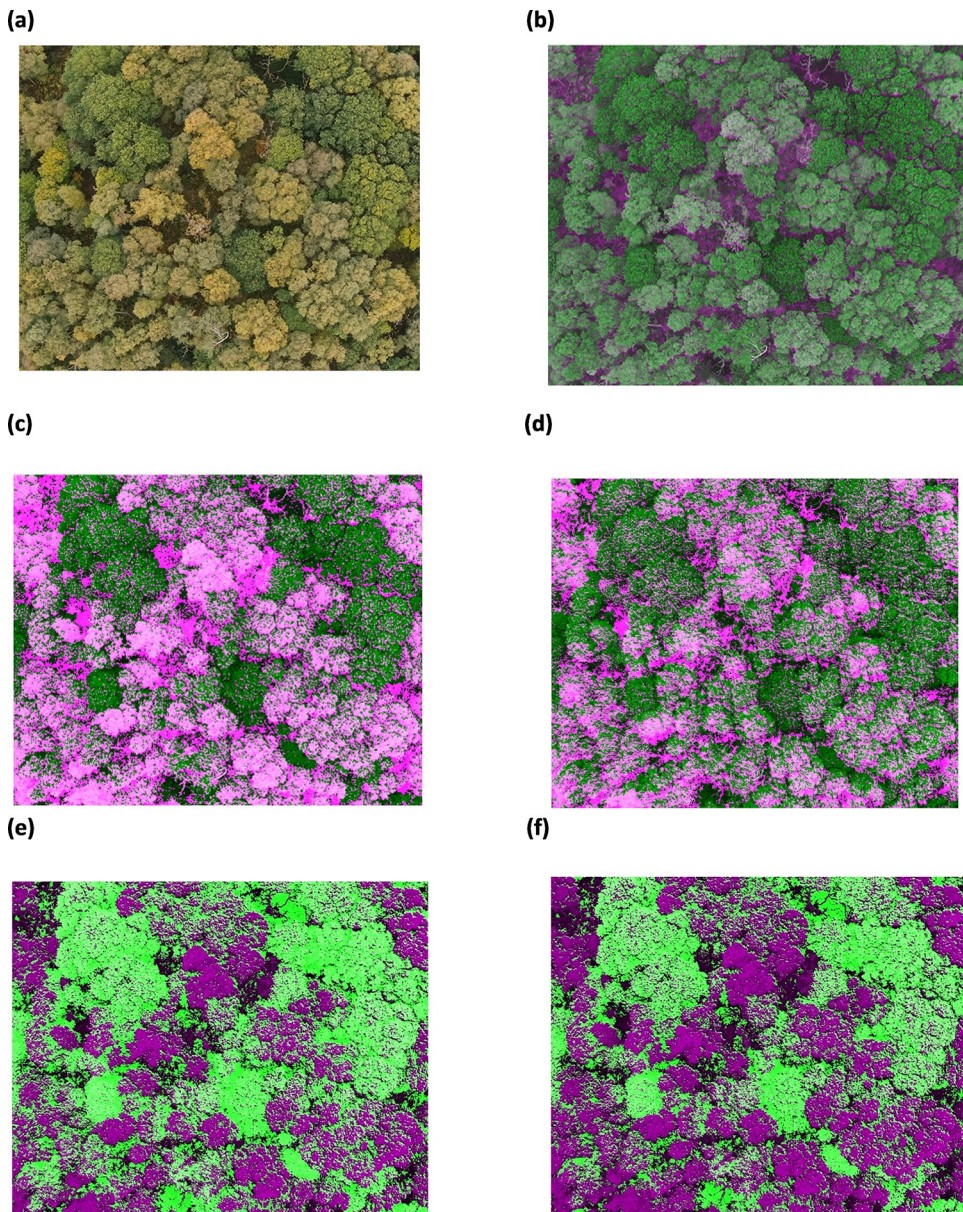

**Fig 9. (a) RGB image (b) new NDSI image differentiating oak and silver birch trees (c) Image (14 October 2020) derived from new NDSI thresholding to segment oak and silver birch trees (d) Image (from 17 September 2020) derived from new NDSI thresholding to segment oak and silver birch trees (e) NDVI image after thresholding (f) PCA derived NDVI image after thresholding.**

segments when compared with the NDVI without weightings from PC1 eigen vectors (Fig 9 (E)). It is noted that the silver birch trees were intensified (in Fig 9(F)) compared to the NDVI image (Fig 9(E)). This clearly demonstrates the significance of the use of PCA classification as the noise from the original multispectral images was reduced, thereby retaining the most important information in each spectral band. While the classification using NDVI with PCA showed possibilities in identifying oak and silver birch trees, its main use has been in its application to represent the vegetation cover between the reflectance energy in red and NIR spectral bands [41].

Therefore, a new spectral index, known as NDSI, was derived by the inclusion and exclusion of specific spectral bands followed by PCA. This novel approach enhances the spectral features which enabled distinguishing between the specific native tree species more precisely. Hence, a new NDSI algorithm was derived using PC1 and PC3 (Fig 7(B)) to identify the tree species followed by further classification using threshold segmentation. The spectral bands of blue, green, red, red edge and NIR showed a huge difference in spectral response between PC1 and PC3 (Fig 7(B)). This difference in spectral response enhances the classification of trees species. Also, the additional spectral bands show a greater sensitivity to the variation of chlorophyll in the tree species which has enabled classification of the tree species where the green regions are segments of oak trees (Fig 9(C)), and the pink regions are segments of silver birch trees (Fig 9(D)). Meanwhile, the NDSI derived using PC1 and PC2 (Fig 7(B)), shows an observable difference in spectral bands between green and red, whereas the remaining spectral bands consist of negative eigenvector components implying that these spectral bands do not hold significant information to help discriminate the native tree species (Fig 8(D)). This further supports the view that the most significant information regarding the use of spectral bands to classify the native tree species of oak and silver birch is retained in PC1 and PC3. This new approach has enabled the identification of the oak trees and the individual standing silver birch trees (Fig 9(C) and 9(D)). This is due to the enhanced spectral resolution observed by inclusion of the additional spectral bands in the new NDSI. This effect has also been reported by Heikkinen et al. [42], who showed that the addition of the red edge band to the existing VIS-NIR band sensors improved the classification of single trees i.e., spruce, pine and silver birch. Additional benefits of the new NDSI are that it has allowed the discrimination of native tree species (Fig 9(C) and 9(D)) which was difficult to observe in the original RGB image (Fig 9(A)). Also, this new approach has proven to provide a finer classification algorithm in comparison to the NDVI algorithm (Fig 9(F)) as the native tree species segments are more distinct (Fig 9(C)). While the new NDSI (Fig 7(B)), using blue, green, red, red edge and NIR spectral bands, could identify the oak and silver birch trees it was not a particularly useful algorithm to identify the dead trees. Therefore, a separate NDSI algorithm was derived using PC1 and PC3 (Fig 7(A)) to identify the dead trees (Fig 10). The new derived NDSI algorithm for identification of dead trees (Fig 10(B)) enhanced the individual branches of the dead trees in comparison to the NDVI (Fig 10(C)) and NDVI applied PCA (Fig 10(D)).

Along with the classification methods of NDVI and PCA, a further useful trait for species classification of the phenology was considered. Phenology is defined as the study of changes in the leaf colour of tree species in deciduous temperate forests in relation to seasonal changes, such as, in autumn; the phenology can also vary amongst different tree species [43]. Due to leaf senescence the decomposition of chlorophyll pigments in different tree species are faster in comparison to anthocyanins and carotenoids [44]. As the images in this study were captured between the 17 September and 14 October 2020, i.e., the onset of autumn, the application of the new NDSI was investigated further. The importance of considering the phenology is exemplified by comparing the data (derived from the data set from 14 October 2020) (Fig 9(B)) with the data set from the 17 September 2020 (Fig 9(C)) which allows better classification of the trees. Further, the spectral response of chlorophyll and carotenoid composition present in trees during phenology indicates a higher spectral response in the blue, red, red edge and NIR spectral bands [45]. Therefore, the new NDSI is composed using these spectral bands, further confirming that UAV multispectral cameras used in this study provide a significant advantage at classifying trees.

The results demonstrate the importance of feature extraction by PCA, which in turn allows selection and elimination of spectral bands according to their spectral response, to derive a new NDSI which when subjected to thresholding, segments the regions of native tree species

(a)

(b)

(c)

(d)

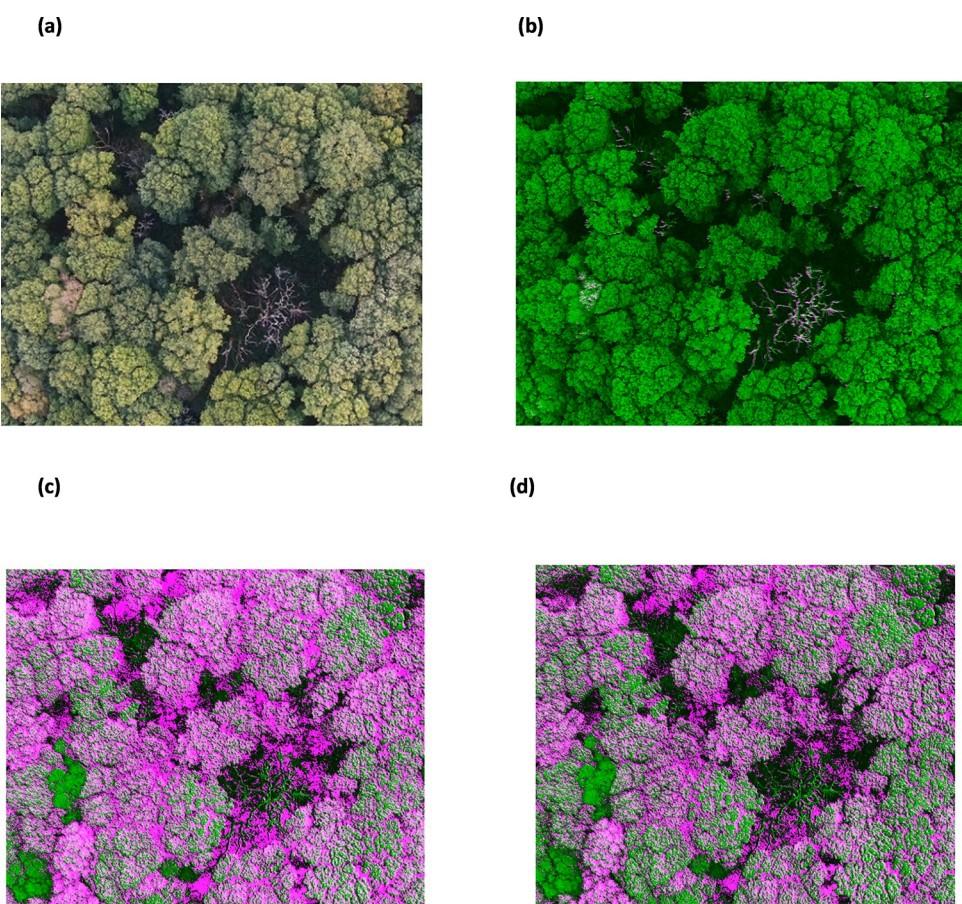

**Fig 10. (a) RGB image outlining dead trees. (b) Image derived from new NDSI to differentiate dead trees after thresholding (14 October 2020) (c) NDVI image after thresholding (d) PCA NDVI image after thresholding.**

for accurate identification (Fig 9(C) & 9(D)). Also, the results highlight the importance of considering the phenological cycle of tree species to aid with their identification using multispectral imagery.

## 3.2 A comparison of quantitative information obtained by UAV MSI and field study data

The new NDSI was used to build orthomosaic images representing the ASNW with native tree species of oak and silver birch trees as well as dead trees. The orthomosaic images were used to count the pixels and calculate the percentage coverage of dead trees and the canopy coverage of oak and silver birch trees on 5 separate occasions (over 4 weeks). The average results of this quantitative results using applied aerial spectroscopy identified 53 ± 1% oak trees and 37 ± 1% silver birch trees. In addition, the aggregated percentage empty space and dead trees was 10.3%; 9 ± 1% empty space and 1 ± 0.15% dead trees, as identified at canopy level (Table 2). It is noted that for a single UAV run data capture was 22 minutes over an area of 7.4 ± 1.3 hectares (73,934 ± 1264 m$^2$). In contrast a detailed ground-level field study was also undertaken over an area of 1.1 hectares (10,800 m$^2$) which resulted in the manual counting and identification of 47% oak trees and 45.6% silver birch trees (Table 2). In addition, manual data gathering was done to identify the percentage dead trees as judged at canopy level (7.4%); this value is

**Table 2. Quantitative information obtained by analysis of UAV MSI and field study data.**

| UAV flight[#] | From analysed UAV data | | | Calculated data | | | | Field study data[@] | | | | | |
|---|---|---|---|---|---|---|---|---|---|---|---|---|---|
| | Total woodland area (m²) | Area of oak tree coverage (m²) | Area of silverbirch tree coverage (m²) | % oak trees | % silverbirch trees | % empty space with lower lying canopy | % dead trees at canopy level | % oak trees | % silverbirch trees | % dead trees at canopy level | | Height of canopy (m)[&] | |
| | | | | | | | | | | Oak | Silverbirch | Oak[$] | Silverbirch[$] |
| 1 | 74,281 | 39,048 | 28,222 | 53 | 38 | 8 | 1.36 | 47.0 | 45.6 | 5.3 | 2.1 | 22 ± 3 | 18 ± 4 |
| 2 | 73,896 | 38,870 | 26,441 | 53 | 36 | 10 | 1.49 | | | | | | |
| 3 | 71,849 | 38,536 | 26,335 | 54 | 37 | 9 | 1.09 | | | | | | |
| 4 | 74,409 | 38,549 | 27,404 | 52 | 37 | 10 | 1.27 | | | | | 22 ± 3 | 18 ± 3 |
| 5 | 75,236 | 39,737 | 27,930 | 53 | 37 | 9 | 1.24 | | | | | | |
| Average | 73,934 | 38,948 | 27,266 | 53 | 37 | 9 | 1.29 | | | | | | |
| SD | 1,264 | 492 | 855 | 1 | 1 | 1 | 0.15 | | | | | | |

Notes.

[#] UAV flights: 1. 17 September 2020; 2. 18 September 2020; 3. 28 September 2020; 4. 8 October 2020; 5. 14 October 2020.

[@] Field data collected on 30 January 2021 over an area of 10,800 m². The sample size was estimated, at the 95% confidence level, with a 5% confidence interval (± margin of error) by assuming every tree could occupy a space of 1 m² (i.e. an assumption that 73,934 trees were present) that the sample size should be 382 trees. In total 453 were manually counted. The field study identified a total of 377 trees as being at canopy level with 7.4% identified as dead trees. Also, an additional 76 trees were identified as dead that were present below canopy level i.e. not visible via the UAV. Additionally, 32 clumps (single or multiple trunk) holly bushes were identified at ground level.

[&] Determined using a hand-held clinometer and calculated using algebra, height was based on two independent people each making repeat measurements (n = 3) on 10 different trees, of the same type, around the field survey site. In addition, the reproducibility was assessed by making repeat measurements on the same tree, by two independent people. Mean height of silverbirch tree (n = 10) as determined by person 1 was 20.35 ± 2.0 m and person 2 was 20.01 ± 2.0 m whereas Mean height of oak tree (n = 10) as determined by person 1 was 18.47 ± 2.0 m and person 2 was 19.83 ± 2.1 m.

[$] mean ± SD, based on the results from two independent field workers.

comparable to the aggregated data obtained using the UAV (10.3%). It was noted at ground level, that a lot of the empty space, at canopy level, was in fact smaller dead trees (Fig 3(B)) and vegetation (e.g. holly bushes, Fig 3(C)). Most dead trees identified in the field study data were below canopy level and appeared blurry or not discernible via UAV data analysis. This phenomena with the remote sensing of species below canopy level has been previously reported [46]. They reported that limitations of data analysis from a UAV platform were influenced by the cloud cover or the angle of the sun at the time of image acquisition and the effect of shadows formed on the species below canopy level from the species above canopy level [46]. The 9% of empty space reported in this study, using the UAV, were identified at ground-level to be due to the presence of, for example, holly bushes and small dead trees (Fig 3). As part of the field study the height of the canopy was estimated, using a hand-held clinometer, to be around 18-22 m. The ground-level field study was done by two people over a period of at least 8 hours, using manual counting, tree identification and marking of the ground using cordoned off areas. It was noted by both approaches that oak trees have broader leaves radiating to branches forming multiple crowns (Fig 9(A)) thereby contributing to a larger canopy of oak in the woodland, and a resultant over-estimation of the number of oak trees using the UAV MSI approach.

## 4. Conclusion

Although there are currently many studies reporting the use of UAVs to identify and quantify tree species, this study has demonstrated some additional benefits. This research has focused

on an area of ASNW to assess both its sustainability and to understand and confirm the native tree species present, particularly the oak and silver birch trees. The derived UAV with MSI results have demonstrated the benefits of a PCA approach which allows the selection of additional spectral bands to derive a new NDSI which assists in classifying tree species. The greater number of spectral bands increases the probability to identify species having different species-specific spectral signatures. Also, thresholding has been discovered as a suitable segmentation method to classify the two types of native tree species. This implies rather than using extensive machine learning methods, simpler discrimination methods of PCA applied NDSI algorithms can be built, and threshold segmentation can be applied to discriminate the tree species. However, to derive a suitable NDSI algorithm it requires subsequent interpretation of data. In addition, the UAV data was compared with a ground level field study that confirmed the accuracy of the newly developed method. Further research will apply this new approach to investigate the transferability of the developed approach to areas of woodland with multiple tree species, as well as its use in precision agriculture.

## Supporting information

**S1 Fig. Stitched orthomosaic images of oak and silver birch trees.**
(DOCX)

**S2 Fig. Stitched orthomosaic images of dead trees.**
(DOCX)

**S1 File.**
(TXT)

## Acknowledgments

We acknowledge the assistance of Jane Young, former ecologist, Northumberland Wildlife Trust, for providing background information on Priestclose Wood. Also, Mr Geoff Dobbins, Senior Estates Officer (Reserves), Northumberland Wildlife Trust Ltd., for permission to fly the UAV over the wood. For field studies we acknowledge the assistance of Lynne Dean and Naomi Dean. In addition, for Priestclose Wood walks, Tom Helm and his boys (Askham, Jess and the late Monty) as well as their friend Harris Dean.

## Author Contributions

**Conceptualization:** Catherine E. Nicholson, Justin J. Perry, John R. Dean.

**Data curation:** Shara Ahmed, Catherine E. Nicholson.

**Formal analysis:** Shara Ahmed, Catherine E. Nicholson.

**Funding acquisition:** Justin J. Perry, John R. Dean.

**Investigation:** Shara Ahmed, Paul Muto, John R. Dean.

**Methodology:** Shara Ahmed, Catherine E. Nicholson, John R. Dean.

**Project administration:** Catherine E. Nicholson, Justin J. Perry, John R. Dean.

**Resources:** Catherine E. Nicholson, John R. Dean.

**Software:** Shara Ahmed, Catherine E. Nicholson.

**Supervision:** Catherine E. Nicholson, Justin J. Perry, John R. Dean.

**Validation:** Shara Ahmed, Catherine E. Nicholson, John R. Dean.

**Visualization:** Shara Ahmed, John R. Dean.

**Writing – original draft:** Shara Ahmed, Catherine E. Nicholson, John R. Dean.

**Writing – review & editing:** Shara Ahmed, Catherine E. Nicholson, Paul Muto, Justin J. Perry, John R. Dean.

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
