## [Decision Letter · Decision Letter 0]

6 Sep 2021

PONE-D-21-23810Applied aerial spectroscopy: A case study on remote sensing of an ancient and semi-natural woodlandPLOS ONE

Dear Dr. Dean,

Thank you for submitting your manuscript to PLOS ONE. After careful consideration, we feel that it has merit but does not fully meet PLOS ONE’s publication criteria as it currently stands. Therefore, we invite you to submit a revised version of the manuscript that addresses the points raised during the review process. The reviewers' considerations must be observed, especially:

- Write about “What is the difference between UAV MSI and Satellite MSI or manned aircraft MSI data”.

- Indicate the gap in the application of NDSI in the UAV MSI in the classification of vegetation species. This can make the new NDSI more valuable.

- Describe how to classify tree species using threshold segmentation (choice of threshold)?

We look forward to receiving your revised manuscript.

Kind regards,

Claudionor Ribeiro da Silva

Academic Editor

PLOS ONE

Journal Requirements:

3. In your Methods section, please provide additional location information, including geographic coordinates for the data set if available.

[No authors have competing interests]. 

5. We note that Figures 1, 2a and 2b in your submission contain map/satellite images which may be copyrighted. All PLOS content is published under the Creative Commons Attribution License (CC BY 4.0), which means that the manuscript, images, and Supporting Information files will be freely available online, and any third party is permitted to access, download, copy, distribute, and use these materials in any way, even commercially, with proper attribution. For these reasons, we cannot publish previously copyrighted maps or satellite images created using proprietary data, such as Google software (Google Maps, Street View, and Earth). For more information, see our copyright guidelines: http://journals.plos.org/plosone/s/licenses-and-copyright.

a) You may seek permission from the original copyright holder of Figures 1, 2a and 2b to publish the content specifically under the CC BY 4.0 license.  

Additional Editor Comments (if provided):

Reviewers' comments:

Reviewer's Responses to Questions

**Comments to the Author**

1. Is the manuscript technically sound, and do the data support the conclusions?

Reviewer #1: Yes

Reviewer #2: Yes

Reviewer #3: Yes

2. Has the statistical analysis been performed appropriately and rigorously? 

Reviewer #1: Yes

Reviewer #2: Yes

Reviewer #3: Yes

3. Have the authors made all data underlying the findings in their manuscript fully available?

Reviewer #1: Yes

Reviewer #2: Yes

Reviewer #3: No

4. Is the manuscript presented in an intelligible fashion and written in standard English?

Reviewer #1: Yes

Reviewer #2: Yes

Reviewer #3: Yes

5. Review Comments to the Author

Reviewer #1: Dear Authors,

Here I would like offer some comments as follows

Minor comments

1. Introduction:

L31-47: Could you reduce description of UAV and increase describe the applicability of UAV MSI with source cite references.

L93-115: I think that you can write about “What is the difference between UAV MSI and Satellite MSI or manned aircraft MSI data”.

L147-167: I did NOT see any application of NDSI from UAV MSI in the previous studies. Could you add these references and state the gap of them in classifying vegetation species? This may make new NDSI more valuable.

2. Results and Discussions

L4187-419: How to classify the tree species using threshold segmentation? Could you describe the threshold choice in detail?

Reviewer #2: The manuscript has chosen a significant subject.The result of this research may be helpful for local authorities to recognize the patterns of future expansion.Now manuscript has resolve all the issues raised.

Reviewer #3: The paper is a concise report of a simple, yet adequate method of identifying trees in a woodland. My suggestion is that the sentence in L146 to 149 needs to be revised. Your attention should focus on how "are required" in L149 fits with the rest of the sentence.

Also, to aid the reproducibility of your work and to conform with the publication policies of PLOS ONE, you should provide a link to the data used for the project?

6. PLOS authors have the option to publish the peer review history of their article (what does this mean?). If published, this will include your full peer review and any attached files.

Reviewer #1: **Yes: **Nguyen Van Trung

Reviewer #2: No

Reviewer #3: **Yes: **Israel Taiwo

---

## [Author Response · Author response to Decision Letter 0]

20 Sep 2021

PONE-D-21-23810

Editor responses:

Dear Claudionor,

We acknowledge and provide the following information.

Methods section: information with regard to the permits for the research have now been included. Previously it was in the acknowledgements.

Competing interests: we have done on the online submission.

Figures 1, 2a and 2b. copyright issues: Figure 1 has been deleted. Figure 2 a and b have been re-done using you direction to the USGS National Map viewer site. The figures in the text have been re-numbered accordingly.

Reviewer 1:

Introduction

Comment on L31-47: Noteworthy recent research of the applicability of multispectral UAV data for species recognition has already been mentioned with references from L128-143.

Comment on L93-115: The main difference between UAV, satellite and manned aircraft data has already been discussed, for example, manned aircraft and satellite data having low temporal resolution therefore cannot be collected multiple times within day (in L101-104). Whereas commentary on the UAV data being collected on regular intervals (in L104-106).

Comment on L146-167: Currently the most used approach, for example, NDVI is used for limited purposes. However, the purpose of the current study was to perform PCA and use the PCA results to derive a new NDSI which helped on the classification of native tree species which has been mentioned (in L146-167).

The main research gap is building up a suitable NDSI by incorporating multiple spectral bands for the specific native tree species to be classified in combination with thresholding method as the currently available method for tree species classification are time consuming requires training data sets as extensive machine learning and deep learning classification models are used. Whereas the method used in the current study is simple, feasible and less time consuming. Also, the newly build NDSI has been compared with NDVI in the results too 

Results and Discussion

Comment on L418-419; The way how thresholding was performed is mentioned in the methodology (in L368-L380).

Reviewer 2:

Thank you.

Reviewer 3:

The sentence has been modified including removal of ‘are required’ (in L150-151).

PLOS Data policy: a link to the raw data has been added.

---

## [Decision Letter · Decision Letter 1]

2 Nov 2021

Applied aerial spectroscopy: A case study on remote sensing of an ancient and semi-natural woodland

PONE-D-21-23810R1

Dear Dr. Dean,

We’re pleased to inform you that your manuscript has been judged scientifically suitable for publication and will be formally accepted for publication once it meets all outstanding technical requirements.

Kind regards,

Claudionor Ribeiro da Silva

Academic Editor

PLOS ONE

Additional Editor Comments (optional):

Reviewers' comments:

Reviewer's Responses to Questions

**Comments to the Author**

1. If the authors have adequately addressed your comments raised in a previous round of review and you feel that this manuscript is now acceptable for publication, you may indicate that here to bypass the “Comments to the Author” section, enter your conflict of interest statement in the “Confidential to Editor” section, and submit your "Accept" recommendation.

Reviewer #1: All comments have been addressed

Reviewer #2: All comments have been addressed

2. Is the manuscript technically sound, and do the data support the conclusions?

Reviewer #1: Yes

Reviewer #2: Yes

3. Has the statistical analysis been performed appropriately and rigorously? 

Reviewer #1: Yes

Reviewer #2: Yes

4. Have the authors made all data underlying the findings in their manuscript fully available?

Reviewer #1: Yes

Reviewer #2: Yes

5. Is the manuscript presented in an intelligible fashion and written in standard English?

Reviewer #1: Yes

Reviewer #2: Yes

6. Review Comments to the Author

Reviewer #1: Dear Authors,

All the comments have been addressed. According to my review, the manuscript could be considered to publish in PLOS ONE.

Thank you

Reviewer #2: All comments is addressed well in revised MS.

The method used in the current research is feasible and less time consuming and NDSI has

been compared with NDVI in the results is also innovative. In my opinion now the revised MS may be accepted for publication.

7. PLOS authors have the option to publish the peer review history of their article (what does this mean?). If published, this will include your full peer review and any attached files.

Reviewer #1: **Yes: **Nguyen Van Trung

Reviewer #2: **Yes: **Dr. Laxmi Kant Sharma, Central University of Rajasthan, India

---

## [Editor Report · Acceptance letter]

5 Nov 2021

PONE-D-21-23810R1 

Applied aerial spectroscopy: A case study on remote sensing of an ancient and semi-natural woodland 

Dear Dr. Dean:

I'm pleased to inform you that your manuscript has been deemed suitable for publication in PLOS ONE. Congratulations! Your manuscript is now with our production department. 

Kind regards, 

on behalf of

Dr. Claudionor Ribeiro da Silva 

Academic Editor

PLOS ONE